# ESTIMATING INSTANCE-DEPENDENT LABEL-NOISE TRANSITION MATRIX USING DNNs

## ABSTRACT

In label-noise learning, estimating the *transition matrix* is a hot topic as the matrix plays an important role in building *statistically consistent classifiers*. Traditionally, the transition from clean labels to noisy labels (i.e., *clean label transition matrix*) has been widely exploited to learn a *clean label classifier* by employing the noisy data. Motivated by that classifiers mostly output *Bayes optimal labels* for prediction, in this paper, we study to directly model the transition from *Bayes optimal labels* to noisy labels (i.e., *Bayes label transition matrix*) and learn a classifier to predict *Bayes optimal labels*. Note that given only noisy data, it is *ill-posed* to estimate either the *clean label transition matrix* or the *Bayes label transition matrix*. But favorably, Bayes optimal labels have less uncertainty compared with the clean labels, i.e., the *class posteriors* of Bayes optimal labels are *one-hot vectors* while those of clean labels are not. This enables two advantages to estimate the *Bayes label transition matrix*, i.e., (a) we could theoretically recover a set of noisy data with Bayes optimal labels under mild conditions; (b) the feasible solution space is much smaller. By exploiting the advantages, we estimate the Bayes label transition matrix by employing a *deep neural network* in a parameterized way, leading to better generalization and superior classification performance.

## 1 INTRODUCTION

The study of classification in the presence of noisy labels has been of interest for three decades (Angluin & Laird, 1988), but becomes more and more important in weakly supervised learning (Thekumparampil et al., 2018; Li et al., 2020b; Guo et al., 2018; Xiao et al., 2015; Zhang et al., 2017a; Yang et al., 2021b;a). The main reason behind this is that datasets are becoming bigger and bigger. To improve annotation efficiency, these large-scale datasets are often collected from crowdsourcing platforms (Yan et al., 2014), online queries (Blum et al., 2003), and image engines (Li et al., 2017), which suffer from unavoidable label noise (Yao et al., 2020a). Recent researches show that the label noise significantly degenerates the performance of deep neural networks, since deep models easily memorize the noisy labels (Zhang et al., 2017a; Yao et al., 2020a).

Generally, the algorithms for combating noisy labels can be categorized into *statistically inconsistent algorithms* and *statistically consistent algorithms*. The statistically inconsistent algorithms are heuristic, such as selecting possible clean examples to train the classifier (Han et al., 2020; Yao et al., 2020a; Yu et al., 2019; Han et al., 2018b; Malach & Shalev-Shwartz, 2017; Ren et al., 2018; Jiang et al., 2018), re-weighting examples to reduce the effect of noisy labels (Ren et al., 2018), correcting labels (Ma et al., 2018; Kremer et al., 2018; Tanaka et al., 2018; Reed et al., 2015), or adding regularization (Han et al., 2018a; Guo et al., 2018; Veit et al., 2017; Vahdat, 2017; Li et al., 2017; 2020b; Wu et al., 2020). These approaches empirically work well, but there is no theoretical guarantee that the learned classifiers can converge to the optimal ones learned from clean data. To address this limitation, algorithms in the second category aim to design *classifier-consistent* algorithms (Yu et al., 2017; Zhang & Sabuncu, 2018; Kremer et al., 2018; Liu & Tao, 2016; Northcutt et al., 2017; Scott, 2015; Natarajan et al., 2013; Goldberger & Ben-Reuven, 2017; Patrini et al., 2017; Thekumparampil et al., 2018; Yu et al., 2018; Liu & Guo, 2020; Xu et al., 2019; Xia et al., 2020b), where classifiers learned on noisy data will *asymptotically converge* to the optimal classifiers defined on the clean domain.

The *label transition matrix* $T(\mathbf{x})$ plays an important role in building *statistically consistent* algorithms. Traditionally, the transition matrix $T(\mathbf{x})$ is defined to relate clean distribution and noisy distribution, where $T(\mathbf{x}) = P(\tilde{Y} \mid Y, X = \mathbf{x})$ and $X$ denotes the random variable of instances/features, $\tilde{Y}$ as the variable for the noisy label, and $Y$ as the variable for the clean label. The above matrix is denoted as the *clean label transition matrix*, which is widely used to learn a *clean label classifier* by employing the noisy data. The learned clean label classifier is expected to predict a probability distribution over a set of pre-defined classes given an input, i.e. *clean class posterior probability* $P(Y \mid X)$. The clean class posterior probability is the distribution from which *clean labels* are sampled. However, *Bayes optimal labels* $Y^*$, i.e., the class labels that maximize the clean class posteriors $Y^* \mid X := \arg\max_Y P(Y \mid X)$, are mostly used as the predicted labels and for computing classification accuracy. Motivated by this, in this paper, we propose to directly model the transition matrix $T^*(\mathbf{x})$ that relates *Bayes optimal distribution* and *noisy distribution*, i.e., $T^*(\mathbf{x}) = P(\tilde{Y} \mid Y^*, X = \mathbf{x})$, where $Y^*$ denotes the variable for *Bayes optimal label*. The *Bayes optimal label classifier* can be learned by exploiting the Bayes label transition matrix directly.

Studying the transition between Bayes optimal distribution and noisy distribution is considered advantageous to that of studying the transition between clean distribution and noisy distribution. The main reason is due to that the *class posteriors* of *Bayes optimal labels* are *one-hot vectors* while those of clean labels are not. Two advantages can be introduced by this to better estimate the instance-dependent transition matrix: *(a) We can collect a set of examples with theoretically guaranteed Bayes optimal labels out of noisy data*.
The intrinsic reason that Bayes optimal labels can be inferred from the noisy data while clean labels cannot is that Bayes optimal labels are the labels that *maximize* the *clean class posteriors* while clean labels are sampled from the *clean class posteriors*. In the presence of label noise, the labels that *maximize* the *noisy class posteriors* could be identical to those that *maximize* the *clean class posteriors* (Bayes optimal labels) under mild conditions (Cheng et al., 2020). Therefore some instances' Bayes optimal labels can be inferred from their *noisy class posteriors* while their clean labels are impossible to infer since the *clean class posteriors* are unobservable, as shown in Figure 1. *(b) The feasible solution space of the Bayes label transition matrix is much smaller than that of the clean label transition matrix.* This is because that Bayes optimal labels have less uncertainty compared with the clean labels. The transition matrix defined by Bayes optimal labels and the noisy labels is therefore *sparse* and can be estimated more efficiently with the same amount of training data.

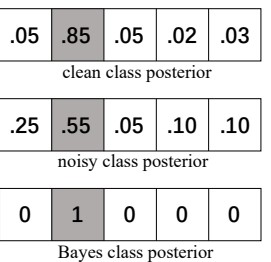

Figure 1: The noisy class posterior is learned from noisy data. Bayes optimal label can be inferred from the noisy class posterior if the noisy rate is controlled. Also, the Bayes optimal label is less uncertain since the Bayes class posterior is *one-hot* vector.

These two advantages naturally motivate us to collect a set of examples with their theoretically guaranteed Bayes optimal labels out of the noisy data to learn to approximate the *Bayes label transition matrix* $T^*(\mathbf{x})$. Due to the high complexity of the instance-dependent matrix $T^*(\mathbf{x})$, we simplify its estimation by parameterizing it using a deep neural network. The collected examples, inferred Bayes optimal labels, and their noisy labels are served as data points to optimize the deep neural network to approximate the $T^*(\mathbf{x})$. Compared with the previous method (Xia et al., 2020a), which made assumptions and leveraged hand-crafted priors to approximate the instance-dependent transition matrices, we train a deep neural network to estimate the *instance-dependent label transition matrix* with a reduced feasible solution space, which achieves lower approximation error, better generalization, and superior classification performance.

## 2 RELATED WORK

**Noise model.** Currently, there are several typical label noise models. Specifically, the random classification noise (RCN) model assumes that clean labels flip randomly with a constant rate (Biggio et al., 2011; Manwani & Sastry, 2013; Natarajan et al., 2013). The class-conditional label noise (CCN) model assumes that the flip rate depends on the latent clean class (Patrini et al., 2017; Xia et al., 2019; Ma et al., 2018). The instance-dependent label noise (IDN) model considers the most general case of label noise, where the flip rate depends on its instance/features (Cheng et al., 2020;

Xia et al., 2020a; Zhu et al., 2020). Obviously, the IDN model is more realistic and applicable. For example, in real-world datasets, an instance whose feature contains less information or is of poor quality may be more prone to be labeled wrongly. The bounded instance dependent label noise (BIDN) (Cheng et al., 2020) is a reasonable extension of IDN, where the flip rates are dependent on instances but upper bounded by a value smaller than 1. However, with only noisy data, it is a *non-trivial* task to model such realistic noise without any assumption (Xia et al., 2020a). This paper focuses on the challenging BIDN problem setting.

**Learning clean distributions.** It is significant to reduce the side effect of noisy labels by inferring clean distributions statistically. The label transition matrix plays an important role in such an inference process, which is used to denote the probabilities that clean labels flip into noisy labels. We first review prior efforts under the class-dependent condition (Patrini et al., 2017). By exploiting the class-dependent transition matrix $T$, the training loss on noisy data can be corrected. The transition matrix $T$ can be estimated in many ways, e.g., by introducing the anchor point assumption (Liu & Tao, 2016), by exploiting clustering (Zhu et al., 2021), by minimizing volume of $T$ (Li et al., 2021), and by using extra clean data (Hendrycks et al., 2018; Shu et al., 2020). To make the estimation more accurately, a slack variable (Xia et al., 2019) or a multiplicative dual $T$ (Yao et al., 2020b) can be introduced to revise the transition matrix. As for the efforts on the instance-dependent transition matrix, existing methods rely on various assumptions, e.g., the noise rate is bounded (Cheng et al., 2020), the noise only depends on the parts of the instance (Xia et al., 2020a), and additional valuable information is available (Berthon et al., 2020). Although the above advanced methods achieve superior performance empirically, the introduction of strong assumptions limit their applications in practice. In this paper, we propose to infer Bayes optimal distribution instead of clean distribution, as Bayes optimal distribution is less uncertain and easy to be inferred under mild conditions.

**Other approaches.** Other methods exist with more sophisticated training frameworks or pipelines, including but not limited to robust loss functions (Zhang & Sabuncu, 2018; Xu et al., 2019; Liu & Guo, 2020), sample selection (Han et al., 2018b; Wang et al., 2019; Lyu & Tsang, 2020), label correction (Tanaka et al., 2018; Zhang et al., 2021; Zheng et al., 2020), (implicit) regularization (Xia et al., 2021; Zhang et al., 2017b; Liu et al., 2020), and semi-supervised learning (Li et al., 2020a; Nguyen et al., 2020).

## 3 PRELIMINARIES

We introduce the problem setting, some important definitions, and the formulation of the proposed *Bayes label transition matrix* in this section.

**Problem setting.** This paper focuses on a classification task given a training dataset with Instance Dependent Noise (IDN), which is denoted by $\tilde{S} = \{(\mathbf{x}_i, \tilde{y}_i)\}_{i=1}^n$. We consider that training examples $\{(\mathbf{x}_i, \tilde{y}_i)\}_{i=1}^n$ are drawn according to random variables $(X, \tilde{Y}) \sim \tilde{\mathcal{D}}$, where $\tilde{\mathcal{D}}$ is a noisy distribution. The noise rate for class $y$ is defined as $\rho_y(\mathbf{x}) = P(\tilde{Y} = y \mid Y \neq y, \mathbf{x})$. This paper focuses on a reasonable IDN setting that the noise rates have upper bounds $\rho_{max}$ as in (Cheng et al., 2020), i.e., $\forall (\mathbf{x}) \in \mathcal{X}, 0 \leq \rho_y(\mathbf{x}) \leq \rho_{max} < 1$. Note that the problem in (Cheng et al., 2020) is defined on *binary classification task* while we extend the problem setting to *multi-class classification*. Our aim is to learn a robust classifier only from the noisy data, which could assign accurate labels for test data.

**Clean distribution.** For the observed noisy training examples, all of them have corresponding clean labels, which are *unobservable*. The clean training examples are denoted by $S = \{(\mathbf{x}_i, y_i)\}_{i=1}^n$, which are considered to be drawn according to random variables $(X, Y) \sim \mathcal{D}$. The term $\mathcal{D}$ denotes the underlying clean distribution.

**Bayes optimal distribution.** Given $X$, its *Bayes optimal label* is denoted by $Y^*$, $Y^* \mid X := \arg\max_Y P(Y \mid X), (X, Y) \sim \mathcal{D}$. The distribution of $(X, Y^*)$ is denoted by $\mathcal{D}^*$. Note the Bayes optimal distribution $\mathcal{D}^*$ is different from the clean distribution $\mathcal{D}$ when $P(Y|X) \notin \{0, 1\}$. Like clean labels, Bayes optimal labels are unobservable due to the information encoded between features and labels is corrupted by label noise (Zhu et al., 2020). Note that it is a *non-trivial task* to infer $\mathcal{D}^*$ only with the noisy training dataset $\tilde{S}$. Also, the noisy label $\tilde{y}$, clean label $y$, and Bayes optimal label $y^*$, for the same instance $\mathbf{x}$ may disagree with each other (Cheng et al., 2020).

**Other definitions.** The classifier is defined as $f : \mathcal{X} \to \mathcal{Y}$, where $\mathcal{X}$ and $\mathcal{Y}$ denote the instance and label spaces respectively. Let $\mathbb{1}[\cdot]$ be the indicator function. Define the *0-1 risk* of $f$ as $\mathbb{1}(f(X), Y) \triangleq \mathbb{1}[f(X) \neq Y]$. Define the *Bayes optimal classifier* $f^*$ as $f^* \triangleq \arg\min_f \mathbb{1}(f(X), Y)$. Note that there is NP-hardness of minimizing the 0-1 risk, which is neither convex nor smooth (Bartlett et al., 2006). We can use the *softmax cross entropy loss* as the *surrogate loss function* to approximately learn the Bayes optimal classifier (Bartlett et al., 2006; Cheng et al., 2020). We aim to learn a classifier $f$ from the noisy distribution $\tilde{\mathcal{D}}$ which also approximately minimizes $\mathbb{E}[\mathbb{1}(f(X), Y)]$.

**Bayes label transition matrix.** Traditional instance-dependent label transition matrix encodes the probabilities that clean labels flip into noisy labels given input instances. However, due to the reasons that clean labels have more uncertainty, estimating the clean label transition matrix is relatively harder. In this paper, we focus on studying the transition from *Bayes optimal labels* to *noisy labels*. We define the Bayes label transition matrix that bridges the Bayes optimal distribution and noisy distribution as follows,

$$T^*_{i,j}(X) = P(\tilde{Y} = j \mid Y^* = i, X), \tag{1}$$

where $T^*_{i,j}(X)$ denotes the $(i, j)$-th element of the matrix $T^*(X)$, indicating the probability of a Bayes optimal label $i$ flipped to noisy label $j$ for input $X$. Given the *noisy class posterior probability* $P(\tilde{\mathbf{Y}} \mid X = \mathbf{x}) = [P(\tilde{Y} = 1 \mid X = \mathbf{x}), \ldots, P(\tilde{Y} = C \mid X = \mathbf{x})]$ (which can be learned from noisy data) and the Bayes label transition matrix $T^*_{ij}(\mathbf{x}) = P(\tilde{Y} = j | Y^* = i, X = \mathbf{x})$, the *Bayes class posterior probability* $P(\mathbf{Y}^* | X = \mathbf{x})$ can be inferred, i.e., $P(\mathbf{Y}^* \mid X = x) = \left( T^*(X = x)^\top \right)^{-1} P(\tilde{\mathbf{Y}} \mid X = x)$.

## 4 METHOD

The feasible solution space of the Bayes label transition matrix is much smaller since Bayes optimal labels are deterministic. Therefore, we propose to estimate a Bayes label transition matrix for each input instance in a parameterized way. We firstly collect a *distilled dataset* (is defined in Defination 1) with theoretically guaranteed Bayes optimal labels out of the noisy dataset (Section 4.1). Then, we can train a deep neural network (Bayes label transition network) on the collected distilled dataset to learn the transition between Bayes optimal labels and noisy labels (Section 4.2). The learned Bayes label transition network is then fixed and used to train a classification network on the noisy dataset in a F-Correction (Patrini et al., 2017) fashion (Section 4.3).

### 4.1 COLLECTING DISTILLED EXAMPLES

In this subsection, we show how to construct a distilled dataset consists of *distilled examples*. We formally introduce the concept of distilled examples first and then present how to collect distilled examples automatically. The collected distilled examples can be used for training the Bayes label transition network.

**Definition 1** (Distilled examples (Cheng et al., 2020)). *An example $(\mathbf{x}, \tilde{y}, y^*)$ is defined to be a distilled example if $y^*$ is identical to the one assigned by the Bayes optimal classifier under the clean data, i.e., $y = f^*_{\mathcal{D}}(\mathbf{x})$.*

The distilled examples can be collected out of noisy examples automatically with the following guarantee,

**Theorem 1** ((Cheng et al., 2020)). *Denote by $\tilde{\eta}_y(\mathbf{x})$ the noisy class posterior probability $P_{\tilde{\mathcal{D}}}(\tilde{Y} = y|X = \mathbf{x})$ and $\eta_y(\mathbf{x}) = P_{\mathcal{D}}(Y = y|X = \mathbf{x})$ the clean class posterior probability. $\forall(x) \in \mathcal{X}$, we have*

$$\tilde{\eta}_y(\mathbf{x}) > \frac{1 + \rho_{max}}{2} \implies \eta_y(\mathbf{x}) > \frac{1}{2} \implies (\mathbf{x}, \tilde{y}, Y^* = y) \text{ is distilled}; \tag{2}$$

where $\rho_{max}$ is the noise rate upper bound. Theorem 1 can be proved in a similar way as did in (Cheng et al., 2020) (Theorem 2 therein). Note the original theorem in (Cheng et al., 2020) was built on *binary-classification* task, we extend it to the *multi-class classification* problem, the extension is straightforward.

According to Theorem 1, we can obtain distilled examples by collecting all noisy examples $(\mathbf{x}, \tilde{y})$ whose $\mathbf{x}$ satisfies $\tilde{\eta}_y(\mathbf{x}) > \frac{1+\rho_{max}}{2}$ and then assigning the label $y$ to it as its Bayes optimal label $y^*$. After that, we obtain an instance of distilled example $(\mathbf{x}^{distilled}, \tilde{y}, \hat{y^*})$, the $\tilde{y}$ indicates the noisy label while the $\hat{y^*}$ indicates the inferred Bayes optimal label. The $\hat{y^*}$ may disagree with the $\tilde{y}$. The conditional probability $\tilde{\eta}_y$ can be estimated as $\hat{\tilde{\eta}}_y$ by several methods, such as the probabilistic classification methods (logistic regression and deep neural networks).

## 4.2 Estimating Bayes label Transition Matrices using Distilled Examples

As discussed in Section 4.1, we can collect a set of distilled examples $(\mathbf{x}^{distilled}, \tilde{y}, \hat{y^*})$ from noisy data. Now we proceed to train a Bayes label transition network parameterized by $\theta$ to estimate the instance-dependent label-noise (IDN) transition matrices, which model the probability of observing a noisy label $\tilde{y}$ given input image $\mathbf{x}$ and its Bayes optimal label $y^*$:

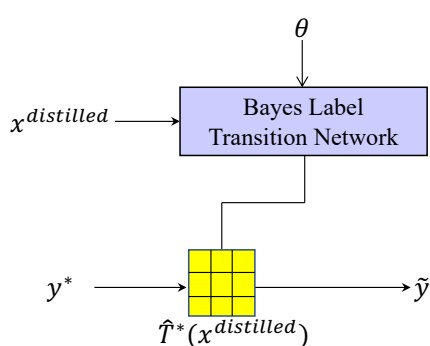

$$\hat{T^*_{i,j}}(\mathbf{x}^{distilled}; \theta) = P(\tilde{Y} = j | Y^* = i, \mathbf{x}^{distilled}; \theta), \tag{3}$$

Specifically, the Bayes label transition network takes $\mathbf{x}^{distilled}$ as input and output an estimated Bayes label transition matrix $\hat{T^*}(\mathbf{x}^{distilled}; \theta) \in \mathbb{R}^{C \times C}$, where $C$ is the number of classes. Note $\hat{T^*}(\mathbf{x}^{distilled}; \theta)$ is an approximation to $T^*(\mathbf{x}^{distilled})$ and it might be not exactly the same as the definition of Bayes label transition matrix. We can use the collected Bayes labels $\hat{y^*}$ and the estimated Bayes label transition matrix $\hat{T^*}(\mathbf{x}_i^{distilled}; \theta)$ to infer the noisy labels. The following empirical risk on the inferred noisy labels and the ground-truth noisy labels are minimized to learn the network's parameter $\theta$:

$$\hat{R}_1(\theta) = -\frac{1}{n} \sum_{i=1}^{n} \tilde{\boldsymbol{y_i}} \log(\hat{\boldsymbol{y_i^*}} \cdot \hat{T^*}(\mathbf{x}_i^{distilled}; \theta)), \tag{4}$$

where $\tilde{\boldsymbol{y_i}}$ and $\hat{\boldsymbol{y_i^*}}$ are $\tilde{y}_i$ and $\hat{y_i^*}$ in the form of *one-hot vectors*, $\tilde{\boldsymbol{y_i}} \in \mathbb{R}^{1 \times C}$ and $\hat{\boldsymbol{y_i^*}} \in \mathbb{R}^{1 \times C}$, respectively. Note that if we have a distilled example for the $i$-th class, we can only make use of it to learn the $i$-th row of the transition matrix. For the other rows, they will not contribute to calculating the loss of the current training example. However, it does not mean that they will be random or not learnable. Their information will be learned by exploiting distilled examples from the non-$i$-th classes. More specifically, the parameters of the network can be divided into row-specific parameters and commonly shared parameters. By assuming that we have distilled examples for each class, both the row-specific parameters and commonly shared parameters will be optimized.

## 4.3 Training Classification Network with Bayes Label Transition Matrices

Our goal is to train a classification network $f(\cdot|w)$ that predicts Bayes class posterior probability $P(Y^* = i | \mathbf{x}; w)$ parameterized by $w$. In the training stage, we cannot observe the Bayes optimal label $Y^*$. Instead, we only have access to noisy label $\tilde{Y}$. The probability of observing a noisy label $\tilde{Y}$ given input image $\mathbf{x}$ is:

$$P(\tilde{Y} = j \mid \mathbf{x}; w, \theta) = \sum_{i=1}^{k} P(\tilde{Y} = j \mid Y^* = i, \mathbf{x}; \theta) P(Y^* = i \mid \mathbf{x}; w), \tag{5}$$

With the trained Bayes label transition network, we can get $\hat{T^*_{i,j}}(\mathbf{x}; \theta) = P(\tilde{Y} = j \mid Y^* = i, \mathbf{x}; \theta)$ for each input $\mathbf{x}$. We exploit F-Correction (Patrini et al., 2017), which is a typical *classifier-consistent* algorithm, to train the classification network. To be specific, fix the learned Bayes label transition network parameter $\theta$, we minimize the empirical risk as follows to optimize the classification network parameter $w$:

$$\hat{R}_2(w) = -\frac{1}{n} \sum_{i=1}^{n} \tilde{\boldsymbol{y_i}} \log(f(\mathbf{x}_i; w) \cdot \hat{T^*}(\mathbf{x}_i; \theta)), \tag{6}$$

---

**Algorithm 1** Bounded Instance-dependent Label Noise Generation.

---

**Require:** Clean examples $\{(\mathbf{x}_i, y_i)\}_{i=1}^n$;
**Require:** Noise rate $\rho$;
**Require:** Noise rate upper bound $\rho_{max}$;
1: Sample instance flip rates $q_i$ from the truncated normal distribution $\mathcal{N}(\rho, 0.1^2, [0, \rho_{max}])$;
   //mean $\gamma$, variance $0.1^2$, range $[0, \rho_{max}]$
2: Independently sample $w_1, w_2, \ldots, w_c$ from the standard normal distribution $\mathcal{N}(0, 1^2)$;
3: **for** $i = 1, 2, \ldots, n$ **do**
4:     $p = \mathbf{x}_i \times w_{y_i}$;    //generate instance-dependent flip rates
5:     $p_{y_i} = -\infty$;    //only consider entries that are different from the true label
6:     $p = q_i \times softmax(p)$;    //make the sum of the off-diagonal entries of the $y_i$-th row to be $q_i$
7:     $p_{y_i} = 1 - q_i$;    //set the diagonal entry to be 1-$q_i$
8:     Randomly choose a label from the label space according to the possibilities $p$ as noisy label $\tilde{y}_i$;
9: **end for**
10: **return** Noisy samples $\{(\mathbf{x}_i, \tilde{y}_i)\}_{i=1}^n$

---

where $f(\mathbf{x}_i; w) \in \mathbb{R}^{1 \times C}$. The F-Correction has been proved to be a classifier-consistent algorithm, the minimizer of $\hat{R}_2(w)$ under the noisy distribution is the same as the minimizer of the original cross-entropy loss under the Bayes optimal distribution (Patrini et al., 2017), if the transition matrix $\hat{T}^*$ is estimated unbiased. Note the Bayes label transition network

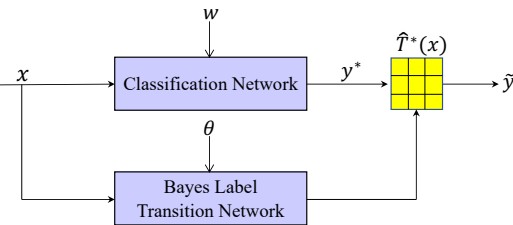

is trained on a biased set, i.e., the set of distilled examples. The network will generalize to the non-distilled examples if they share the same pattern with the distilled examples which causes label noise. A recent study (Xia et al., 2020a) empirically verified that the patterns that cause label noise are commonly shared. Our empirical experiments further show that the network $\hat{T}^*(\mathbf{x}; \theta)$ generalizes well to unseen examples and thus helps achieve superior classification performance.

## 5 EXPERIMENTS

In this section, we first introduce the experiment setup (Section 5.1) including the datasets used (Section 5.1.1) and the compared methods (Section 5.1.2). Next, we analyse the hyper-parameter sensitivity in Section 5.2. Finally, we present and analyze the experimental results on synthetic and real-world noisy datasets to show the effectiveness of the proposed method (Section 5.3). The implementation details and more ablation studies are included in the Appendix.

### 5.1 EXPERIMENT SETUP

In this section, we introduce the datasets we used and the baseline methods we compared with.

#### 5.1.1 DATASETS

We conduct the experiment on four datasets to verify the effectiveness of our proposed method, where three of them are manually corrupted, i.e., *F-MNIST*, *CIFAR-10*, and *SVHN*, one of them is real-world noisy datasets, i.e., *Clothing1M*. *F-MNIST* has $28 \times 28$ grayscale images of 10 classes including 60,000 training images and 10,000 test images. *CIFAR-10* dataset contains 50,000 color images from 10 classes for training and 10,000 color images from 10 classes for testing both with shape of $32 \times 32 \times 3$. *SVHN* has 10 classes of images with 73,257 training images and 26,032 test images. We manually corrupt the three datasets, i.e., *F-MNIST*, *CIFAR-10* and *SVHN* with bounded instance-dependent label noise according to Algorithm 1, which is modified from (Xia et al., 2020a). In noise generation, the noise rate upper bound $\rho_{max}$ in Algorithm 1 is set as 0.6 for all experiments. All experiments on those datasets with synthetic instance-dependent label noise are repeated five times to guarantee reliability. The *Clothing1M* has 1M images with real-world noisy labels for

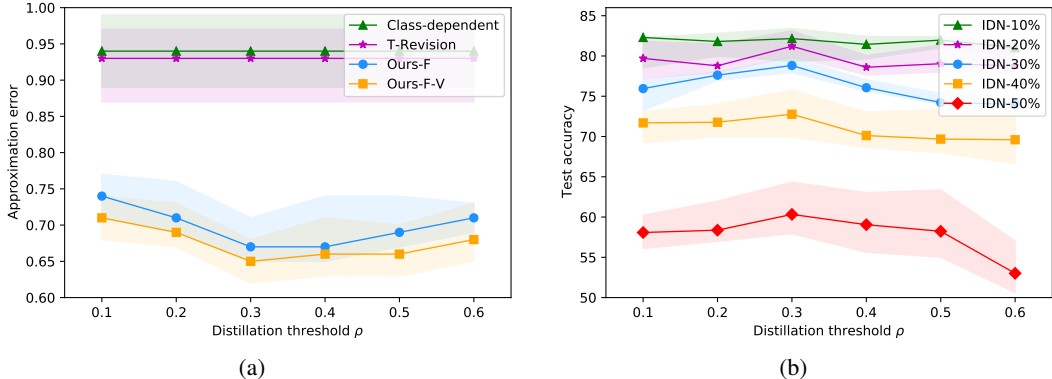

Figure 2: Illustration of the transition matrix approximation error and the hyperparameter sensitivity. Figure (a) illustrates how the distillation threshold $\rho$ affects the approximation error for the instance-dependent transition matrix. Figure (b) illustrates how the distillation threshold $\rho$ affects the test classification performance. The error bar for standard deviation in each figure has been shaded.

training and 10k images with the clean label for testing. 10% of the noisy training examples of all datasets are left out as a noisy validation set for model selection.

### 5.1.2 COMPARISON METHODS

We compare the proposed method with several state-of-the-art approaches: (1) CE, which trains the classification network with the standard cross-entropy loss on noise datasets. (2) GCE (Zhang & Sabuncu, 2018), which unites the mean absolute error loss and the cross-entropy loss to combat noisy labels. (3) APL (Ma et al., 2020), which combines two mutually reinforcing robust loss functions, we employ its combination of NCE and RCE for comparison. (4) Decoupling (Malach & Shalev-Shwartz, 2017), which trains two networks on samples whose predictions from two networks are different. (5) MentorNet (Jiang et al., 2018), Co-teaching (Han et al., 2018b), and Co-teaching+ (Yu et al., 2019) mainly handle noisy labels by training networks on instances with small loss values. (6) Joint (Tanaka et al., 2018), which jointly optimizes the network parameters and the sample labels. The hyperparameters $\alpha$ and $\beta$ are set to 1.2 and 0.8, respectively. (7) DMI (Xu et al., 2019), which proposes a novel information-theoretic loss function for training neural networks robust to label noise. (8) Forward (Patrini et al., 2017), Reweight (Liu & Tao, 2016), and T-Revision (Xia et al., 2019) utilize a class-dependent transition matrix $T$ to correct the loss function. (9) PTD (Xia et al., 2020a), estimates instance-dependent transition matrix by combing part-dependent transition matrices, which is the most related work to our proposed method. We also provide comparison results between our method and DivideMix(Li et al., 2020a), which is a hybrid algorithm that combines multiple powerful techniques, e.g. Gaussian Mixture Model, MixMatch, MixUp, regularization and asymmetric noise penalty in Appendix. As for our method, we simply model the instance-dependent matrix by employing a neural network.

### 5.2 HYPER-PARAMETER SENSITIVITY

The quality of the distilled dataset relies on the choice of distillation threshold $\hat{\rho}_{max}$ (denoted as $\hat{\rho}$ in the following paragraph) in Theorem 1. To further explore the effect of $\hat{\rho}$, we conduct hyper-parameter sensitivity studies on *CIFAR-10* in this section.

In Figure 2(a), we show the instance-dependent transition matrix approximation error when employing the class-dependent transition matrix, the revised class-dependent transition matrix, and our proposed instance-dependent transition matrix estimation method. The error is measured by $\ell_1$ norm between the ground-truth transition matrix and the estimated transition matrix. For each instance, we only analyze the approximation error of a specific row because the noisy label is generated by one row of the instance-dependent transition matrix. The "Class-dependent" represents the class-dependent transition matrix learning methods (Patrini et al., 2017), the 'T-Revision' indicates the class-dependent transition matrix is revised by a learnable slack variable (Xia et al., 2019). Our pro-

Table 1: Means and standard deviations (percentage) of classification accuracy on *F-MNIST* with different label noise levels. '-V' indicates matrix revision (Xia et al., 2019).

|  | IDN-10% | IDN-20% | IDN-30% | IDN-40% | IDN-50% |
|---|---|---|---|---|---|
| CE | 88.65 ± 0.45 | 88.31 ± 0.37 | 85.22 ± 0.56 | 76.56 ± 2.50 | 67.42 ± 3.91 |
| GCE | 90.86 ± 0.38 | 88.59 ± 0.26 | 86.64 ± 0.76 | 76.93 ± 1.64 | 66.69 ± 1.07 |
| APL | 86.46 ± 0.27 | 85.32 ± 0.88 | 85.59 ± 0.85 | 74.66 ± 2.77 | 62.82 ± 0.44 |
| Decoupling | 89.83 ± 0.45 | 86.29 ± 1.13 | 86.01 ± 1.01 | 78.78 ± 0.53 | 67.33 ± 1.33 |
| MentorNet | 90.35 ± 0.64 | 87.92 ± 0.83 | 87.24 ± 0.99 | 79.01 ± 2.30 | 66.44 ± 2.97 |
| Co-teaching | 90.65 ± 0.58 | 88.77 ± 0.41 | 86.98 ± 0.67 | 78.92 ± 1.36 | 67.66 ± 2.42 |
| Co-teaching+ | 90.47 ± 0.98 | 89.15 ± 1.77 | 86.15 ± 1.04 | 79.23 ± 1.30 | 63.49 ± 2.94 |
| Joint | 80.19 ± 0.99 | 78.46 ± 1.24 | 72.73 ± 2.44 | 65.93 ± 2.08 | 50.93 ± 3.52 |
| DMI | 91.58 ± 0.46 | 90.33 ± 0.66 | 85.96 ± 1.52 | 77.77 ± 2.15 | 68.02 ± 1.59 |
| Forward | 89.65 ± 0.24 | 88.61 ± 0.77 | 85.01 ± 0.43 | 78.59 ± 0.38 | 67.11 ± 1.46 |
| Reweight | 90.33 ± 0.27 | 88.81 ± 0.44 | 84.93 ± 0.42 | 76.07 ± 1.93 | 67.66 ± 1.65 |
| S2E | 91.04 ± 0.92 | 89.93 ± 1.08 | 86.77 ± 1.15 | 76.12 ± 1.21 | 70.24 ± 2.64 |
| T-Revision | 91.36 ± 0.59 | 90.24 ± 1.01 | 85.59 ± 1.77 | 78.24 ± 1.12 | 69.04 ± 2.92 |
| PTD | 92.03 ± 0.33 | 90.78 ± 0.64 | 87.86 ± 0.78 | 79.46 ± 1.58 | 73.38 ± 2.25 |
| Ours | 96.06 ± 0.71 | 94.97 ± 0.33 | 91.47 ± 1.36 | 82.88 ± 2.72 | 76.35 ± 3.79 |
| Ours-V | **96.93 ± 0.31** | **95.55 ± 0.59** | **92.24 ± 1.87** | **83.43 ± 1.72** | **76.89 ± 4.26** |

Table 2: Means and standard deviations (percentage) of classification accuracy on *CIFAR-10* with different label noise levels. '-V' indicates matrix revision (Xia et al., 2019).

|  | IDN-10% | IDN-20% | IDN-30% | IDN-40% | IDN-50% |
|---|---|---|---|---|---|
| CE | 73.54 ± 0.14 | 71.49 ± 1.35 | 67.52 ± 1.68 | 58.63 ± 4.92 | 51.54 ± 2.70 |
| GCE | 74.24 ± 0.89 | 72.11 ± 0.43 | 69.31 ± 0.18 | 56.86 ± 0.92 | 53.44 ± 1.28 |
| APL | 71.12 ± 0.19 | 68.89 ± 0.27 | 65.17 ± 0.35 | 53.22 ± 2.21 | 47.31 ± 1.41 |
| Decoupling | 73.91 ± 0.37 | 74.23 ± 1.18 | 70.85 ± 1.88 | 54.73 ± 1.02 | 52.04 ± 2.09 |
| MentorNet | 74.93 ± 1.37 | 73.59 ± 1.29 | 72.32 ± 1.04 | 57.85 ± 1.88 | 52.96 ± 1.98 |
| Co-teaching | 75.49 ± 0.47 | 75.93 ± 0.87 | 74.86 ± 0.42 | 59.07 ± 1.03 | 55.62 ± 3.93 |
| Co-teaching+ | 74.77 ± 0.16 | 75.14 ± 0.61 | 71.92 ± 2.13 | 59.15 ± 0.87 | 53.02 ± 3.34 |
| Joint | 75.97 ± 0.98 | 76.45 ± 0.45 | 75.93 ± 1.65 | 63.22 ± 5.37 | 55.84 ± 3.25 |
| DMI | 74.65 ± 0.13 | 73.49 ± 0.88 | 73.93 ± 0.34 | 60.22 ± 3.47 | 54.35 ± 2.28 |
| Forward | 72.35 ± 0.91 | 70.98 ± 0.32 | 66.53 ± 1.96 | 58.63 ± 1.25 | 52.33 ± 1.65 |
| Reweight | 73.55 ± 0.32 | 71.49 ± 0.57 | 68.76 ± 0.37 | 60.32 ± 1.03 | 52.03 ± 1.70 |
| S2E | 75.93 ± 1.01 | 75.53 ± 0.32 | 71.21 ± 2.51 | 64.62 ± 0.68 | 56.03 ± 1.07 |
| T-Revision | 74.01 ± 0.45 | 73.42 ± 0.64 | 71.15 ± 0.43 | 59.93 ± 1.33 | 55.67 ± 2.07 |
| PTD | 76.33 ± 0.38 | 76.05 ± 1.72 | 75.42 ± 1.33 | 65.92 ± 2.33 | 56.63 ± 1.88 |
| Ours | 81.73 ± 0.56 | 80.26 ± 0.63 | 77.69 ± 1.37 | 71.96 ± 2.27 | 59.15 ± 3.11 |
| Ours-V | 82.16 ± 1.01 | 80.37 ± 1.98 | 78.82 ± 1.07 | **72.93 ± 4.00** | **60.33 ± 5.29** |

posed method estimates an instance-dependent transition matrix for each input. It can be observed that our proposed method can achieve a much lower approximation error. Figure 2(b) shows the classification performance of our proposed method when choosing various distillation threshold $\hat{\rho}$s. When $\hat{\rho}$ is not too large or too small, our method is not sensitive to the choice of $\hat{\rho}$. More experimental results on the quality of distilled dataset when applying different values of $\hat{\rho}$ are included in the appendix B. We manually set $\hat{\rho} = 0.3$, a decent trade-off between distillation accuracy and the number of distilled examples, in all experiments later to avoid laborious hyper-parameter tuning and access to the true noise rate.

## 5.3 COMPARISON WITH THE STATE-OF-THE-ARTS

**Results on synthetic noisy datasets.** Table 1,2 and 3 report the classification accuracy on the datasets of *F-MNIST*,*CIFAR-10*, and *SVHN*, respectively.

For *F-MNIST*, our method surpasses all the baseline methods by a large margin. Equipping the transition matrix revision (-V) (Xia et al., 2019) can further boost the performance of our method.

Table 3: Means and standard deviations (percentage) of classification accuracy on *SVHN* with different label noise levels. '-V' indicates matrix revision (Xia et al., 2019).

|  | IDN-10% | IDN-20% | IDN-30% | IDN-40% | IDN-50% |
|---|---|---|---|---|---|
| CE | $90.39 \pm 0.13$ | $89.04 \pm 1.32$ | $85.65 \pm 1.84$ | $79.94 \pm 2.71$ | $61.01 \pm 5.41$ |
| GCE | $90.82 \pm 0.15$ | $89.35 \pm 0.94$ | $86.43 \pm 0.63$ | $81.66 \pm 1.58$ | $54.77 \pm 0.25$ |
| APL | $71.78 \pm 0.76$ | $89.48 \pm 1.67$ | $83.46 \pm 2.17$ | $77.90 \pm 2.31$ | $55.25 \pm 3.77$ |
| Decoupling | $90.55 \pm 0.83$ | $88.74 \pm 0.77$ | $85.03 \pm 1.63$ | $83.36 \pm 2.73$ | $56.76 \pm 1.87$ |
| MentorNet | $90.28 \pm 0.52$ | $89.09 \pm 0.95$ | $85.89 \pm 0.73$ | $82.63 \pm 1.73$ | $55.27 \pm 4.14$ |
| Co-teaching | $91.05 \pm 0.33$ | $89.56 \pm 1.77$ | $87.75 \pm 1.37$ | $84.92 \pm 1.59$ | $59.56 \pm 2.34$ |
| Co-teaching+ | $92.83 \pm 0.87$ | $90.73 \pm 1.39$ | $86.37 \pm 1.66$ | $75.24 \pm 3.77$ | $54.58 \pm 3.46$ |
| Joint | $88.39 \pm 0.62$ | $85.37 \pm 0.44$ | $81.56 \pm 0.43$ | $78.98 \pm 2.98$ | $59.14 \pm 3.22$ |
| DMI | $92.11 \pm 0.49$ | $91.63 \pm 0.87$ | $86.98 \pm 0.36$ | $81.11 \pm 0.68$ | $63.22 \pm 3.97$ |
| Forward | $90.01 \pm 0.78$ | $89.77 \pm 1.54$ | $86.70 \pm 1.44$ | $80.24 \pm 2.77$ | $57.57 \pm 1.45$ |
| Reweight | $91.06 \pm 0.19$ | $92.01 \pm 1.04$ | $87.55 \pm 1.71$ | $83.79 \pm 1.11$ | $55.08 \pm 1.25$ |
| S2E | $92.70 \pm 0.51$ | $92.02 \pm 1.54$ | $88.77 \pm 1.77$ | $83.06 \pm 2.19$ | $65.39 \pm 2.77$ |
| T-Revision | $93.07 \pm 0.79$ | $92.67 \pm 0.88$ | $88.49 \pm 1.44$ | $82.43 \pm 1.77$ | $67.64 \pm 2.57$ |
| PTD | $93.77 \pm 0.33$ | $92.59 \pm 1.07$ | $89.64 \pm 1.98$ | $83.56 \pm 2.21$ | $71.57 \pm 3.32$ |
| Ours | $96.05 \pm 0.32$ | $94.97 \pm 0.58$ | $93.99 \pm 1.24$ | $87.67 \pm 1.29$ | $78.13 \pm 4.62$ |
| Ours-V | $\mathbf{96.37 \pm 0.77}$ | $95.12 \pm 0.40$ | $\mathbf{94.69 \pm 0.24}$ | $\mathbf{88.13 \pm 3.23}$ | $\mathbf{78.71 \pm 4.37}$ |

For *SVHN* and *CIFAR-10*, the superiority of our method is gradually revealed along with the noise rate increase, which shows that our method can handle the extremely hard situation much better. Specifically, the classification accuracy of our method is 5.83% higher than PTD (the best statistically consistent baseline) on *CIFAR-10* in the IDN-10% case, and the performance gap is enlarged to 7.01% in the IDN-40% case. On the *SVHN*, the classification accuracy of our method is 2.60% higher than PTD in the IDN-10% case, 5.05% higher than PTD in the IDN-30% case, and 7.14% higher than PTD in the most challenging IDN-50% case.

### 5.3.1 RESULTS ON REAL-WORLD DATASETS

The noise model of real-world datasets is more likely to be instance-dependent. Our proposed method also performs favorably on the challenging *Clothing1M* dataset (Table 4), which proves that our method is more flexible to handle such real-world noise problem .

Table 4: Classification accuracy on *Clothing1M*. In the experiments, only noisy samples are exploited to train and validate the deep model.

| CE | Decoupling | MentorNet | Co-teaching | Co-teaching+ | Joint | DMI |
|---|---|---|---|---|---|---|
| 68.88 | 54.53 | 56.79 | 60.15 | 65.15 | 70.88 | 70.12 |

| Forward | Reweight | T-Revision | PTD | PTD-V | Ours | Ours-V |
|---|---|---|---|---|---|---|
| 69.91 | 70.40 | 70.97 | 70.07 | 70.26 | **73.33** | **73.39** |

## 6 CONCLUSION

In this paper, we focus on training the robust classifier with the challenging instance-dependent label noise. To address the issues of existing *clean label transition matrix*, we propose to directly build the transition between *Bayes optimal labels* and *noisy labels*. By reducing the feasible solution of the transition matrix estimation, we prove that the instance-dependent label transition matrix that relates *Bayes optimal labels* and *noisy labels* can be directly learned using *deep neural networks*. The main limitation of our method comes from that the *distilled examples* are collected out of noisy data leading to unavoidable data distribution bias to the transition matrix estimation. Experimental results demonstrate that the proposed method is more superior in dealing with instance-dependent label noise, especially for the case of high-level noise rates.

## 7 ETHICS STATEMENT

This paper doesn't raise any ethics concerns. This study doesn't involve any human subjects, practices to data set releases, potentially harmful insights, methodologies and applications, pontential conflicts of interest and sponsorship, discrimination/bias/fairness concerns, privacy and security issues, legal compliance, and research integrity issues.

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

## A    IMPLEMENTATION DETAILS

We use ResNet-18 (He et al., 2016) for *F-MNIST*, ResNet-34 networks (He et al., 2016) for *CIFAR-10* and *SVHN*. We first use SGD with momentum 0.9, batch size 128, and an initial learning rate of 0.01 to warm up the network for five epochs on the noisy dataset. For *Clothing1M*, we use a ResNet50 pretrained on ImageNet, and the learning rate is set as 1e-3. Then, we use the warm-upped network to collect distilled examples from noisy datasets according to Section 4.1. After distilled examples collection, we train the instance-dependent transition matrix estimator network on the distilled dataset for 5 epochs. The Bayes label transition network is the same architecture as the classification network, but the last linear layer is modified according to the transition matrix shape. The optimizer of the Bayes label transition network is SGD, with a momentum of 0.9 and a learning rate of 0.01. Then, we fix the trained Bayes label transition network to train the classification network. The Bayes label transition network is used to generate a transition matrix for each input image; the transition matrix is used to correct the outputs of the classification network to bridge the Bayes posterior and the noisy posterior. The classification network is trained on the noisy dataset for 50 epochs for *F-MNIST*, *CIFAR-10* and *SVHN* and for 10 epochs for *Clothing1M* using Adam optimizer with a learning rate of $5e-7$ and weight decay of $1e-4$. We also apply the transition matrix revision technique (Xia et al., 2019) to boost the performance. Note for a fair comparison, we do not use any data augmentation technique in all experiments as in (Xia et al., 2020a). All the codes are implemented in PyTorch 1.6.0 with CUDA 10.0, and run on NVIDIA Tesla V100 GPUs.

## B    THE QUALITY OF DATASET DISTILLATION

The distillation threshold $\hat{\rho}$ controls how many examples can be collected out of noisy dataset and the quality of the distilled dataset. We analyse the effect of $\hat{\rho}$ on the CIFAR-10 dataset in Table. 5. The distillation accuracy is computed by counting how many inferred Bayes optimal labels are consistent with their corresponding clean labels among all distilled examples.

| Noise rate | $\hat{\rho} = 0.3$ | | $\hat{\rho} = 0.5$ | |
|---|---|---|---|---|
| | distill. acc. | # of distilled examples | distill. acc. | # of distilled examples |
| IDN-10% | 98% | 27983 / 50000 | 99% | 19983 / 50000 |
| IDN-30% | 96% | 17673 / 50000 | 99% | 10673 / 50000 |
| IDN-50%. | 94% | 8029 / 50000 | 98% | 5098 / 50000 |

Table 5: Distillation quality analysis on CIFAR-10, with total 50,000 examples in the original non-distilled dataset.

## C    ABLATION ON BAYES LABEL TRANSITION MATRIX

To verify the effectiveness of the estimated Bayes label transition matrix, we compare our method with some ablated variants, e.g. directly train a classifier on the distilled dataset and relabel the noisy dataset using the classifier trained on distilled dataset.

| | CIFAR-10 IDN-10% | CIFAR-10 IDN-30% | Clothing1M |
|---|---|---|---|
| Training classifier on distilled dataset | 74.56 | 67.42 | 62.37 |
| Relabeling noisy dataset | 76.68 | 70.73 | 64.98 |
| Ours | **82.16** | **78.82** | **73.39** |

## D    COMPARISION WITH DIVIDEMIX

DivideMix has a much more complicated pipeline than us and is not a statistically consistent algorithm. We compare our method with DivideMix to further show the effectiveness and flexibility of our proposed method. Compared with DivideMix, our method exhibit competitive performance when noise rate is low and surpass DivideMix by a large margin on the worst noise cases (3.22%

performance improvement on CIFAR-10 and 4.38% on SVHN, both under IDN-50% ), with a much simpler and flexible algorithm design.

|  | IDN-10% | IDN-20% | IDN-30% | IDN-40% | IDN-50% |
|---|---|---|---|---|---|
| DivideMix | $96.37 \pm 0.72$ | $95.92 \pm 0.73$ | $90.37 \pm 0.83$ | $80.92 \pm 2.32$ | $74.63 \pm 3.76$ |
| Ours | $\mathbf{96.93 \pm 0.31}$ | $\mathbf{95.55 \pm 0.59}$ | $\mathbf{92.24 \pm 1.87}$ | $\mathbf{83.43 \pm 1.72}$ | $\mathbf{76.89 \pm 4.26}$ |

Table 6: F-MNIST

|  | IDN-10% | IDN-20% | IDN-30% | IDN-40% | IDN-50% |
|---|---|---|---|---|---|
| DivideMix | $\mathbf{83.31 \pm 0.23}$ | $\mathbf{81.42 \pm 0.28}$ | $\mathbf{80.73 \pm 1.28}$ | $70.29 \pm 1.97$ | $57.11 \pm 3.64$ |
| Ours | $82.16 \pm 1.01$ | $80.37 \pm 1.98$ | $78.82 \pm 1.07$ | $\mathbf{72.93 \pm 4.00}$ | $\mathbf{60.33 \pm 5.29}$ |

Table 7: CIFAR10

|  | IDN-10% | IDN-20% | IDN-30% | IDN-40% | IDN-50% |
|---|---|---|---|---|---|
| DivideMix | $96.02 \pm 0.45$ | $\mathbf{95.73 \pm 0.48}$ | $92.07 \pm 1.47$ | $85.69 \pm 2.47$ | $74.33 \pm 4.07$ |
| Ours | $\mathbf{96.37 \pm 0.77}$ | $95.12 \pm 0.40$ | $\mathbf{94.69 \pm 0.24}$ | $\mathbf{88.13 \pm 3.23}$ | $\mathbf{78.71 \pm 4.37}$ |

Table 8: SVHN

