# OpenReview forum: "Estimating Instance-dependent Label-noise Transition Matrix using DNNs"
_ICLR.cc/2022/Conference — ICLR 2022 Submitted_

### Official Review · Reviewer_ieRv · 2021-11-01

**Correctness:** 3
**Technical Novelty And Significance:** 3
**Empirical Novelty And Significance:** 3
**Recommendation:** 5
**Confidence:** 4

**Main Review:**

Strength:

(1) The view of dealing with Bayes class posterior instead of clean class posterior is new, and seems provides benefits.

(2) Explicitly parameterizing the transition matrix is good, and might have advantages in task generalization consideration.


Weakness:

(1) No theoretical guarantee that dealing with Bayes class posterior transition is better than conventional noise transition matrix estimation methods, either in robustness or generalization error.

(2) It seems that final method is not intrinsically related with Bayes optimal labels, except that soft labels were replaced by hard ones.

(3) Experiments were only conducted on datasets with relatively few classes, such as CIFAR-10  and SVHN with only 10 classes.

**Summary Of The Paper:**

This paper proposed a new noise-transition-matrix based label correction method for robust deep learning against label noise. Different from conventional methods, that estimating noise transition matrix from clean class posterior to the noisy one, the proposed method try to estimation the transition matrix from Bayes class posterior to the noisy one, which is expected to reduce the feasible solution space, and lead to better performance.

**Summary Of The Review:**

The idea is new and interesting, but final solution is not satisfactory, especially for theoretical guarantee.

---

> ### Author Response · Authors · 2021-11-25
> **Response to Reviewer ieRv**
>
> Dear Reviewer,
>
> Thanks for your comments! Regarding the theoretical guarantee, we would like to clarify a bit. The contribution of the paper is not about theory, but about a new concept and a new algorithm. Specifically, given only noisy data, the traditional instance-dependent label noise transition matrix (a random walk from instances with clean labels to noisy labels) is not identifiable and is hard to learn. Very little work has done to model the real-world label noise. Motivated by the theory in Cheng et al. (2020), we conceptually propose a new transition matrix (a random walk from instances with Bayes optimal labels to noisy labels), which is identifiable for some instances, making it possible to estimate the instance-dependent label noise learnable from only noisy data. This is also empirically verified by the new algorithm. We do believe the contribution will interest many readers in the community of learning with noisy labels.
>
> Best,
>
> Authors

---

### Official Review · Reviewer_M4SW · 2021-11-01

**Correctness:** 2
**Technical Novelty And Significance:** 1
**Empirical Novelty And Significance:** 2
**Recommendation:** 3
**Confidence:** 5

**Main Review:**

This paper mainly applies the idea in Cheng et al. (2020) to multiclass problems to identify "distilled examples". The extension is straightforward, but is problematic. The use of *Bayes label transition matrix* seems novel, but lacking justifications make it a minor novelty. Finally, using DNNs for estimations has nothing new. Overall this paper is not novel nor significant. The correctness and clarity are also of concern.

---

Strengths:

- The use of *Bayes label transition matrix* to learn from instance-dependent noisy labels seems novel.
- The experiments seem comprehensive.

---

Weaknesses:

1. The idea to identify "distilled examples" is neither novel nor significant. This paper simply applies the setting and the idea in Cheng et al. (2020), the bounded instance-dependent label noise paper. Although the authors claimed they extended Cheng et al. (2020) from binary to multiclass, this extension is neither novel nor significant. Instead, it is quite problematic. First, Cheng et al. (2020) already has showed that their framework can be easily extended to multiclass setting in the supplementary material. Second, in this paper, the authors used the same theorem (Theorem 1) to identify distilled examples as that in Cheng et al. (2020). This is problematic: although it is true that $\eta_y(x) > 0.5$ implies $y$ is the Bayes label in multiclass, this is only a sufficient condition. In fact, requiring $\eta_y(x) > 0.5$ is the so-called "dominant label" condition [1], and is generally considered as a quite strong assumption. Third, Cheng et al. (2020) also considered covariate shift and the case when $\rho_{max}$ is not known, but such extensions are missing in this work. To summarize, this work only trivially applies some of Cheng et al. (2020)'s results to multiclass to identify "distilled examples". This is not a meaningful contribution.

2. The use of *Bayes label transition matrix* seems novel, but lacks justifications. The paper claims (second paragraph in page 2, (b)) that "The feasible solution space of the Bayes label transition matrix is much smaller than that of the clean label transition matrix.", but no justifications are given for the claimed "sparsity". Recall that "sparsity" means that entries of a matrix are mostly zeros. Such claim is not true for the *Bayes label transition matrix*, even if the Bayes class posterior is sparse. In Section 4.2, the authors acknowledged that "if we have a distilled example for the i-th class, we can only make use of it to learn the i-th row of the transition matrix.", but the explanation for why the other rows are not random nor learnable is far from satisfactory. More theoretical and empirical work is needed to understand and justify the use of *Bayes label transition matrix*.

3. The use of DNNs to estimate *Bayes label transition matrix* has nothing new. No insights about how to design such nets are provided.

4. The noise generating process (Algorithm 1) does not reflect the bounded noise assumption. At the beginning of Section 3, it is assumed that $\forall x, 0 \le \rho_y(x) \le \rho_{max} < 1$, where $\rho_y(x)  = P(\tilde{Y} = y | Y \ne y, x)$. However, in Algorithm 1, it is only guaranteed that for each $i$, $P(\tilde{Y} \ne y_i | Y = y_i, x_i) \le \rho_{max}$. Therefore, the proposed noise generating process does not satisfy the bounded noise assumption.

5. In experiments, specifically Section 5.2, the baseline does not make sense, because both "class-dependent" and "T-Revision" are for CCN setting (the label noise only depends on the label, but not on the instance). The authors should have compared with methods that are designed for instance-dependent label noise.

6. The clarity of this paper is of concern. The writing quality is below the requirements of this conference. For example, the paper used a long paragraph in page 1 to discuss about *statistical consistency*, but such information is not directly related to this work since this work does not have consistency results. BTW, Xia et al. (2020b) is not classifier-consistent.

---

Other minor issues:

1. In Figure 1, the noisy class posterior does not sum to 1.

---
Refs:

[1] Tong Zhang:
Statistical Analysis of Some Multi-Category Large Margin Classification Methods. J. Mach. Learn. Res. 5: 1225-1251 (2004)

**Summary Of The Paper:**

This work proposes to use *Bayes label transition matrix* (instead of the common *clean label transition matrix*) to learn from instance-dependent noisy labels. The Bayes label transition matrix is estimated by DNNs.

**Summary Of The Review:**

This paper has no novel and significant contribution. The correctness and clarity are also of serious concern. Therefore I vote for a reject.

---

> ### Author Response · Authors · 2021-11-25
> **To Reviewer M4SW**
>
> ### 1 & 2 & 3:
>
> We have **never** claimed the extension of Cheng et al. (2020) from binary classification to multi-class classification is one of our contributions. We have made it **very clear** in the paper that we just borrowed it from Cheng et al. (2020) to infer Bayes labels.
>
> We would like to **emphasize again that the main contribution of this work** is, from the philosophical perspective, this work is **the first to provide a general framework that successfully estimates the instance-dependent label-noise transition matrix in a parametric way, and empirically performs significantly better than previous non-parametric label-noise matrix estimation baselines.**
>
> We **do not agree with you** that estimating the instance-dependent label-noise transition matrix using DNNs is nothing new. Since the instance-dependent transition matrix is **unique** for each input instance, it makes much sense that estimate it in a parametric way. **To the best of our knowledge, our paper is the first one that successfully estimate the instance-dependent transition matrix using DNNs.** Can you give some references to support your claim **nothing new**?
>
> To estimate the transition matrix by only employing noisy data is a key research problem in label-noise learning. Class-dependent label noise transition matrix has been extensively studied but not the instance-dependent ones (which is unique for each input instance). To address this, we propose a conceptually new definition of instance-dependent label noise transition matrix. For the first time, we enable the instance-dependent transition matrix to be identifiable for the instances whose Bayes optimal labels can be identified. (Before our work, the instance-dependent transition matrix is only identifiable to anchor points, whose clean label is of one-hot vectors). Accordingly, we also propose a new DNN-based algorithm to estimate it. We do believe the contribution will interest many readers in the community of label-noise learning.
>
> ### 4:
> In our experiments, for all datasets, the number of training samples is uniformly distributed across classes, and we use the same noise rate for all classes. Therefore, algorithm 1 satisfied the bounded noise rate assumption.
>
> ### 5 & 6:
> Thanks and we will improve the clarity.

---

> > ### Comment · Reviewer_M4SW · 2021-11-29
> > **Changed my score to 3.**
> >
> > Dear authors,
> >
> > Thanks for your response. My apologies if my review was too harsh and made you unhappy. I agree that the paper has a good idea, but unfortunately it suffered from many issues (the overall flow and the writing, the experimental design, and some errors), which make the paper below the acceptance requirement.
> >
> > Re. Cheng et al. (2020), in Section 4.1, just below Theorem 1, you emphasized the difference between Cheng et al. and this paper. My understanding was that you tried to make it sound like a contribution (as also pointed out by Reviewer dRUQ). Again, my main issue with Section 4.1 is that the you just used Cheng et al., and as I have pointed out, such trivial extension to multiclass is very problematic: the resulting criterion to find distilled examples in multiclass is way too strong. Since distilled examples are critical in the estimation of the Bayes label transition matrix, this weakens the results.
> >
> > Re. "nothing new", I meant "no insights about how to design such nets are provided", because using DNNs for estimation tasks is really not a new idea generally speaking. Plus, [2] also used neural nets to estimate instance-dependent transition matrix (a different parametric form).
> >
> > Please also double check #4. I believe either the noise generating process (Algorithm 1) does not reflect the bounded noise assumption, or the assumption is not correctly specified.
> >
> > To summarize, I have tried my best understanding this paper and providing detailed constructive suggestions for improvements. Unfortunately I could not give higher score for this paper in its current form. Addressing the issues above could make this paper much stronger.
> >
> > ---
> >
> > Ref:
> >
> > [2] Part-dependent Label Noise: Towards Instance-dependent Label Noise
> > Xiaobo Xia, Tongliang Liu, Bo Han, Nannan Wang, Mingming Gong, Haifeng Liu, Gang Niu, Dacheng Tao, Masashi Sugiyama

---

### Official Review · Reviewer_6uGG · 2021-11-02

**Correctness:** 4
**Technical Novelty And Significance:** 4
**Empirical Novelty And Significance:** 3
**Recommendation:** 8
**Confidence:** 4

**Main Review:**

Strengths:

This paper transforms the original clean transition matrix estimation problem to the estimation of transition from Bayes optimal labels to noisy labels, which is quite interesting and novel for the noisy label learning.  A DNN is used to estimate the transition matrix, which can be optimized simultaneously with the classifier in an end-to-end manner. This is also a major contribution in this area. Extensive experiments are conducted to demonstrate the method. And the results have shown great improvements. This can be considered as another highlight of this paper.

Minor weaknesses:

The authors introduced a parametric Bayes label transition network, which is a DNN. Thus, the authors should compare the extra parameter quantity of the proposed method with those of related methods.

It would be good if the author can provide an example of the learned transition matrix from Bayes optimal labels to noisy labels. Or there should be some investigation about how the Bayes-transition matrix was being estimated, empirically or theoretically.







**Summary Of The Paper:**

The transition matrix plays a vital role in modeling label noise. Current methods focus on modeling the transition from clean labels to noisy labels. While this paper alternatively models the transition from Bayes optimal labels to noisy labels. Since we usually use the Bayes optimal labels for prediction. This transformation will not affect the practical use but makes the estimation of the matrix much easier.  Specifically, this paper designs a DNN to estimate the transition matrix. During training, the DNN can be optimized with the classifier simultaneously in an end-to-end manner.  Extensive experiments are conducted to support the proposed method.

**Summary Of The Review:**

The method has made certain theoretical contributions to the label noise learning community.  The algorithm also clearly outperforms the current SotA. Therefore, I would like to lean on the positive side of this paper.

---

> ### Author Response · Authors · 2021-11-25
> **Response to Reviewer 6uGG**
>
> Dear Reviewer,
>
> Thanks for your comments. Regarding the parameter quantity, at the training stage, we leverage an extra network to capture the image patterns and generate instance-dependent label-noise transition matrices. The estimated matrices are used to correct the training loss of the classification network. **It is worth noting that our method doesn’t involve any extra parameters/memory cost during the inference/test stage**, because only the classification network is used to make predictions during the test. Compared to other works, e.g. co-teaching [1] and decoupling [2] which leverage two networks during both training and test, our method relies on fewer computing resources during inference.
>
> |   |  Number of parameters during training | Number of parameters during test  |
> | :------------: | :------------: | :------------: |
> |  Ours |  2N |  N |
> |  co-teaching [1] | 2N  |  2N |
> |  decoupling [2] | 2N  |  2N |
>
> where 'N' indicates the number of parameters of a classification network.
>
> [1] Bo Han et al. Co-teaching: Robust Training of Deep Neural Networks with Extremely Noisy Labels. NeurIPS 2018.
> [2] Eran Malach et al. Decoupling "when to update" from "how to update". NeurIPS 2017.

---

### Official Review · Reviewer_dRUQ · 2021-11-02

**Correctness:** 2
**Technical Novelty And Significance:** 2
**Empirical Novelty And Significance:** 4
**Recommendation:** 3
**Confidence:** 4

**Main Review:**

To the best of my knowledge, this is the first paper that introduces to model Bayes label transition rather than clean label transition. The paper is well written and easy to follow. Results are very promising compared to other SOTA methods.

My main concern is on the novelty of the contribution. Given that the idea of modelling Bayes label transition is a new and interesting idea, the techniques are mostly based on existing works:
- Step 1: The Bayes label collection closely follows the framework proposed by Cheng et al. (2020). Note that the extension to multiclass setting, despite being straightforward as agreed by the authors, was also presented in the Supplementary Section B in Cheng et al. (2020) already.
- Step 2: By replacing clean labels with Bayes labels, the authors model transition matrix with a DNN. This seems to be the main contribution (as it is included in the paper title too?) but in experiments the authors simply did straightforward adaptation of standard DNN architecture for this task. Some ablation or other experiments to compare to non-parametric models for (Bayes) label transition could also support the paper.
- Step 3: The final classification algorithm uses F-correction proposed by Patrini et al. (2017).

Given the relatively weak methodological contribution, the current paper is also a bit weak in terms of theoretical contributions. For instance,
- As stated in Sec 1, one of the motivations for modelling Bayes label transition is that Bayes label has a smaller feature solution space hence easier to model because it is one-hot, but it would support the paper if the authors can include a formal definition and/or theoretical justification on this claim.
- Besides, I would argue that Bayes label transition being easier to model does not mean that using Bayes labels to learn a weakly supervised model is preferred to using inferred clean labels in practice, unless the impact of error propagation from label transition to weakly supervised model is carefully studied. Again theoretical justification may strongly improve the paper.
- In Sec 4.3, the authors acknowledged that the Bayes transition estimation through distilled examples breaks the assumption that F-correction requires an unbiased estimate of the transition matrix, but then only briefly mentioned that the network can generalise well to non-distilled examples empirically. This is only a claim at best, and also breaks the motivation that the authors are looking for classifier-consistent algorithms.

Other minor concerns/suggestions:
- Since the paper significantly relies on the distilled example collection from Cheng et al. (2020), Cheng et al. (2020) spent quite some space discussing the known issues and mitigation strategies of their distilled example collection (e.g., the automatically-collected distilled examples only is not statistically consistent, covariate shift correction, how to collect without knowledge on noise rate bound). The current paper only has ablation on the impact of noise rate bound (Sec 5.2), it would be worth discussing the impact of any other known issues of their distilled example collection on the currently proposed methods.

**Summary Of The Paper:**

The paper proposes to estimate an Instance-Dependent Noise (IDN) label transition matrix. Instead of modelling the clean label transition as typically done in previous literature, the authors propose to estimate the Bayes label transition using a DNN, motivate by several advantages including theoretically guaranteed Bayes label collection and smaller feasible solution space, hence empirically easier to model. Controlled experiments show consistent improvement over other SOTA methods in noisy label transition.

**Summary Of The Review:**

This paper introduces an interesting idea to model Bayes label transition rather than clean label transition based on several existing literatures. However, the overall contribution of the paper is a bit weak for ICLR standards IMHO, and some theoretical justification are currently missing to support most of their claims.

---

> ### Author Response · Authors · 2021-11-25
> **Response to Reviewer dRUQ**
>
> Thanks for your comments. We **strongly suggest one can evaluate this work based on the whole framework rather than decompose it into multiple components and try to find related works for each part**. Actually, from the philosophical perspective, **this work is the first one** that explicitly model the transition matrix from Bayesian labels to noisy labels. It is also **the first one**  that provide a general framework for estimating the instance-dependent label-noise transition matrix in a parametric way, and empirically performs significantly better than previous non-parametric label-noise matrix estimation baselines.
>
> For the dataset distillation (Theorem 1), we never claim it as one of our contributions. In contrast, we have made it clear that we just borrowed it and provided sufficient citation and discussion about Cheng et al. (2020).
>
> Thanks,
>
> Authors.

---

> > ### Comment · Reviewer_dRUQ · 2021-11-29
> > **I'm keeping my scores.**
> >
> > Thanks for the authors to respond to my concerns!
> >
> > First of all, I would like to apologize if any of my reviews offended the authors. I was not intending to decompose the proposed method, but rather to understand the significance of the contributions. In the end, I see now that the authors are re-iterating that, as a whole, the novelty is a new pipeline that uses a combination of existing methods. In that case I'd re-position to judge the paper as **a good empirical study**, so I prefer to keep my scores as "not good enough" since I can hardly recommend a higher score for the current version of the paper. I do agree that the empirical results seem strong, but as a reader, I'm afraid I'm not fully aware where the improvement comes from after reading the paper (e.g. from which step or must I use the proposed pipeline as a whole? Is it perhaps due to parametric over non-parametric estimation of transition?)
> >
> > Besides, I'd strongly suggest that significant changes to the writing are needed, as from the introduction it looked like the authors were trying to propose a method motivated by strong theories, but I was did not see any novel theories nor a thorough understanding of the proposed methods, as several other reviewers seemed to have also concerns about.
> >
> > Regarding the extension to a multiclass setting, I think the confusion may have come from the fact that the authors stressed several times in writing that "(Cheng et al., 2020) built on binary-classification task" and "we extend it to the multi-class classification problem" as a contribution though "the extension is straightforward". Again, changing the writing can make clear the main contribution of the paper.

---

> > > ### Author Response · Authors · 2021-11-30
> > > **Thank the reviewer**
> > >
> > > We would like to thank the reviewer. We value the suggestions and will take them in the next version.

---

### Decision · Program_Chairs · 2022-01-20

**Decision:**

Reject

**Comment:**

This paper received a majority voting of rejection. In the internal discussion, one reviewer updated his/her score from 1 to 3 according to the author response. I have read all the materials of this paper including manuscript, appendix, comments and response. Based on collected information from all reviewers and my personal judgement, I can make the recommendation on this paper, *rejection*. Here are the comments that I summarized, which include my opinion and evidence.

**Interesting Idea**

Every reviewer including me agree that the idea of modelling Bayes label transition is novel and interesting.

**The motivation lacks of supportive evidence**

The second motivation that "the feasible solution space of the Bayes label transition matrix is much smaller than that of the clean label transition matrix" is not well supported. The authors should theoretically or empirically demonstrate this point. The current description on uncertainty is not strong enough. Moreover, if so, the benefits are not illustrated. The feasible solution space, even with a small coverage area is continuous with infinite solutions.

**A new concept**

The authors tried to sell the concept of a new transition matrix, but failed. I believe it might result from the organization and presentation. The authors spent too much pages introducing others' work. At least, a formal definition of the new concept should be given. In the current version, Definition 1 is from Cheng et al., 2020 on distilled examples.

**Title**

Literally from title, I guess DNN is a key component or a selling point of this paper. Actually no. We expect the authors could provide the insights on what benefits are using DNN over other techniques and how to apply DNN to estimate the transition matrix. If this is not a selling point, this word might be removed from the title.

**Algorithm 1**

I am a little surprised that the only algorithm listed in this paper is label noise generation. Instead the proposed algorithm of this paper is expected.

**Experimental Evaluation**

The experimental results look much better than other baselines. It is a little confusing that some best results are bold, some not.

**Presentation**

Although I did not notice obvious grammar errors, some sentences are very long (3 lines). They made difficulties to follow the idea. I have to read these sentences several times. In my eyes, this is the biggest one! Presentation means how to sell the idea to audience (not only reviewers, but also future readers) in an easy understood way. The current version spent much space introducing others' work; on the contrary, the original or key part is not well illustrated.

Although this paper has a novel idea and good experimental support, other issues listed above demonstrate the current version is not ready for a top-tier conference. No objection from reviewers was raised to again this recommendation.